# A “Green” Homogenate Extraction Coupled with UHPLC-MS for the Rapid Determination of Diterpenoids in *Croton Crassifolius*

**DOI:** 10.3390/molecules24040694

**Published:** 2019-02-15

**Authors:** Jin-Long Tian, Chi Shu, Ye Zhang, Hui-Jun Cui, Xu Xie, Xu-Long Ran, Tian-Shun Chen, Zhi-Huan Zang, Jian-Guo Liu, Bin Li

**Affiliations:** College of Food Science, Shenyang Agricultural University, Shenyang 110866, China; sweet_vs_sweet@163.com (J.-L.T.); 2018500024@syau.edu.cn (C.S.); zysyau@163.com (Y.Z.); chj0423@163.com (H.-J.C.); syauxiexu@163.com (X.X.); ranxulong12@163.com (X.-L.R.); cts166282@163.com (T.-S.C.); m139983676611@163.com (Z.-H.Z.); 18547552365@163.com (J.-G.L.)

**Keywords:** *Croton crassifolius*, clerodane diterpenoids, homogenate extraction, UHPLC–MS, principle component analysis

## Abstract

Clerodane diterpenoids are the main bioactive constituents of *Croton crassifolius* and are proved to have multiple biological activities. However, quality control (QC) research on the constituents are rare. Thus, the major research purpose of the current study was to establish an efficient homogenate extraction (HGE) process combined with a sensitive and specific ultra-high-performance liquid chromatography-tandem mass spectrometry (UHPLC–MS) technique together for the rapid extraction and determination of clerodane diterpenoids in *C. crassifolius*. All calibration curves showed good linearity (*r* > 0.9943) within the test ranges and the intra- and inter-day precisions and repeatability were all within required limits. This modified HGE–UHPLC–MS method only took 5 min to extract nine clerodane diterpenoids in *C. crassifolius* and another 12 min to quantify these components. The results indicated that the quantitative analysis based on UHPLC–MS was a feasible method for QC of clerodane diterpenoids in *C. crassifolius*, and the findings outlined in the current study also inferred the potential of the method in the QC of clerodane diterpenoids in other complex species of plants.

## 1. Introduction

*Croton crassifolius* (Euphorbiaceae, Croton) is a herb employed in Traditional Chinese Medicine (TCM) and mainly grows in Thailand, Laos, Vietnam, and Southern China [1,2]. The herb is also known as “ji-gu-xiang” and its roots are employed for the treatments of cancer, rheumatism, sore throat, and stomachache for centuries [3,4]. Phytochemical studies have suggested that clerodane diterpenoids are the main bioactive constituents of *C. crassifolius*, which are responsible for its antiviral, cytotoxic, and anti-angiogenic activities [1,4,5,6]. However, even with the promising application potential, few studies regarding the effective pre-treatment and quality control (QC) of clerodane diterpenoids have been performed in *C. crassifolius*. Thus, it is reasonable to explore effective methods for the efficient extraction, separation, and analysis of clerodane diterpenoids in *C. crassifolius*.

Currently, ultrasonic-assisted extraction (UAE), heating reflux extraction (HRE), and infusion extraction (IE) are widely used to extract diterpenoids from *C. crassifolius* [7], but all these methods have some limitations such as low yield and long extraction time. Of the different methods, homogenate extraction (HGE) has shown its potential to effectively extract chemical constituents in plants to solvents through high-speed crushing, fluid cutting, mixing, and mechanical shearing without pressure or heating. In addition, dust pollution (dust generated by crushing Traditional Chinese Medicine before extraction) can be avoided in HGE combining grinding and extraction [8,9,10,11].

To our knowledge, only HPLC-DAD (high performance liquid chromatography-diode array detector) method was developed for quantitative assay of six diterpenoid in *C. crassifolius* [7] and no systematic research has been carried out on them in plants. Compared to the HPLC–DAD methods, the UHPLC–MS methods have the advantage of a higher sensitivity, higher selectivity, and higher throughput for quantitation [12,13,14]. Therefore, an in-depth study of clerodane diterpenoids by UHPLC–MS methods in *C. crassifolius* are of great importance.

In the present study, an efficient, sensitive, and selective HGE–UHPLC–MS method was proposed to extract and quantify nine clerodane diterpenoids (Figure 1) in *C. crassifolius*. The UHPLC-MS method combined with principal component analysis (PCA) is of great value for the QC of *C. crassifolius* from different production regions. In addition, the method can be applied for the extraction and determination of clerodane diterpenoids in other complex species of plants.

## 2. Results

### 2.1. Clerodane Diterpenoids and Anti-Angiogenic Activity

The isolation of compounds **1–9** from *C. crassifolius* was implemented (Appendix A), and ^1^H and ^13^C NMR spectroscopy were used to confirm their identities (Appendix A). The effect of diterpenoids **1–9** on vascular endothelial growth Factor (VEGF) release is shown in Appendix A.

### 2.2. HGE–UHPLC–MS Method to Rapid Determination of Nine Diterpenoids

#### 2.2.1. Extraction Procedure

The effect of the extraction time (3, 5, 7, 9, 11 min), solvent-to-solid ratio (10, 20, 30, 40, 50 mL/g), ethanol concentration (0, 30%, 50%, 60%, 70%, 80%, 90%), and voltage (50, 60, 70, 80, 90 V) were assessed to realize the optimization of the HGE procedure (Figure 2). Through the above study, the optimal HGE conditions to extract the nine diterpenoids in *C. crassifolius* were known with the solvent-to-solid ratio of 40 mL/g, the extraction time of 5 min, 70% ethanol as the ideal extraction solvent, and an extraction voltage of 80 V. And the extraction yields (contents of nine diterpenoids **1**–**9**) were determined by the methods described in Section 3.5. UHPLC and MS Conditions.

#### 2.2.2. Optimization of UHPLC-MS Conditions

The target diterpenoids have a narrow polarity range (structural similarity), and *C. crassifolius* contains the constituents with high complexity. Therefore, the desired resolution was obtained by using gradient elution. A comparison was carried out between methanol-water and acetonitrile-water mobile phase systems. The separation ability of acetonitrile-water was better for the diterpenoids. The three chromatographic columns (a Waters CORTECS C_18_ (100 × 2.1 mm, 1.6 μm), a Waters ACQUITY UPLC BEH Shield RP18 (100 × 2.1 mm, 1.7 μm), and a Waters ACQUITY UPLC HSS T3 (100 × 2.1 mm, 1.7 μm)) were compared to get the best resolution of diterpenoids through the Waters CORTECS C_18_ column (100 × 2.1 mm, 1.6 μm), with shorter chromatographic separation times and symmetric peak shapes. The MS detection sensitivity was improved and the peak tailing was decreased by adding formic acid and acetic acid with different concentrations to the aqueous phase. The results of 0.1% formic acid were the best. In summary, the optimal mobile phase contained A (0.1% *v*/*v* formic acid/acetonitrile) and B (0.1% *v*/*v* formic acid/water) (Figure 3).

To optimize the MS conditions, negative and positive ion mode tests were implemented for MS analysis. Compared with negative mode with stable and intense molecular ion peaks, the nine diterpenoids had higher sensitivity in the positive mode and cleaner mass spectral backgrounds, [M + H]^+^ for **3** (*m*/*z* 343.0) and **6** (*m*/*z* 357.0), [M + Na]^+^ for **5** (*m*/*z* 439.2), **7** (*m*/*z* 351.1), **8** (*m*/*z* 379.1), and **9** (*m*/*z* 397.2), and [M + NH_4_]^+^ for **2** (*m*/*z* 390.0) and **4** (*m*/*z* 346.0). To optimize the multiple reaction monitoring (MRM) conditions, the precursor ion relative abundance was obtained, and the ions were produced to optimize the parameters of fragmentor voltage and collision energy. Table 1 showed the final conditions of collision energy and fragmentor voltage.

#### 2.2.3. Method Validation

##### Calibration Curve, Limit of Detection (LOD), and Limit of Quantification (LOQ)

In a wide range of concentration, analytes had high linearity in the concentration range (*r* > 0.9943). The LOQs and LODs were lower than 4.78 ng/mL and 0.924 ng/mL, respectively. The results were shown in Table 2.

##### Precision

The intra- and inter-day precision values were evaluated through the relative standard deviations (RSDs) of peak areas, which were smaller than 4.54 and 5.96% (Table 2), respectively. Therefore, for quantitative analysis, the UHPLC–MS method was sensitive and precise.

##### Recovery

The recovery (96.3–104.5%) of all analytes was high in the fortified samples at three spiking levels, and RSDs were lower than 4.21%. Table 2 summarizes the data, indicating the accurate measurement for the nine analytes.

##### Stability and Reproducibility

As shown in Table 2, the sample solution had satisfactory stability at 4 °C. After running the reproducibility (*n* = 5) samples, the RSD (%) values were calculated. The results obtained with RSD of peak areas lower than 5.71% were satisfactory, indicating the chromatographic separation had good reproducibility.

#### 2.2.4. Sample Analysis

The nine major bioactive diterpenoids in different samples of *C. crassifoliu* obtained from provinces in Southern China were analyzed through the validated HGE-UHPLC-MS method. Table 3 shows the contents of the investigated nine diterpenoids. The contents of the nine analytes ranged from 31.10 to 318.59 mg/g, demonstrating the samples had different contents of the nine diterpenoids. The variations resulted from both extrinsic factors, such as climatic or geographic conditions, harvest time, storage conditions, and herb processing, and intrinsic factors, such as plant origin and genetic variations. Moreover, the results of quantitative analysis indicated that diterpenoids were found in high concentrations in *C. crassifolius* and there were marked differences in the content of the nine diterpenoids in *C. crassifolius*. In the majority of cases, compound **4** was the dominant component, and its content was much higher than that other diterpenoids. The contents of compounds **5** and **6** were relatively low, and compound **8** was not present or below the limit of detection in some samples. However, several active components, including micro- and macro-components, are responsible for the therapeutic effects. Therefore, it is reasonable to analyze multiple components to control the quality of *C. crassifolius*.

#### 2.2.5. Principle Component Analysis

Although there were clear differences in the content of various sample sources, the sources could hardly be distinguished. The differences or similarity of multivariate data can be visualized through PCA as unsupervised multivariate-data analysis [15]. In this study, PCA was used to distinguish the 12 batches of *C. crassifolius* and samples were well separated by this PCA method. In the scatter plot (Figure 4A), samples were classified into three groups, labelled groups I-III, including group I (Yunnan-1 and Yunnan-2), group II (Guangdong-1, Guangdong-2, and Guangdong-3), and group III (Guangxi-1, Guangxi-2, Guangxi-3, Hainan-1, Hainan-2, Guizhou-1, and Guizhou-2). In general, loading plots provided useful information to identify important features in the first and second PC dimensions [16]. The results showed (Figure 4B) the contributions of different variables to sample differentiation, and the principal component contents of the points were similar (compounds **4**, **6**, **8**, and **9**). Variables opposite to each other were negatively correlated (compounds **2**/**3** and **5**), and the variables located 90 degrees from each other were almost uncorrelated (compounds **3, 5**/**1, 7** and **2**/**1, 7**). Compounds **6**–**9** had high scores for the principal components, showing the significant relationship between the compounds and sample variations. According to the PCA results, different batches of *C. crassifolius* had different chemical compositions. The quantitative analysis based on UHPLC–MS can assess and control the quality of *C. crassifolius*.

## 3. Materials and Methods

### 3.1. Chemicals and Reagents

In our laboratory, after isolating and purifying compounds **1–9** from *C. crassifolius*, ^1^H and ^13^C NMR spectroscopy were used to confirm their identity (Appendix A). The peak areas which were detected by UHPLC–MS were normalized, and the compounds had a purity higher than 98%. The UHPLC–MS was analyzed using HPLC-grade acetonitrile (Fisher, Waltham, MA, USA). The HPLC grade formic acid (50% in water) was obtained from Sigma–Aldrich (St. Louis, MO, USA). A Milli-Q system (Millipore, Burlington, MA, USA) was employed to purify the ultra-pure water. The extraction was implemented through analytical-grade ethanol (Tianjin Chemical Corporation, Tianjin, China).

### 3.2. Anti-Angiogenic Activity Assay

The anti-angiogenic activity with enzyme-linked immunosorbent assay (ELISA) was evaluated by the VEGF content in HepG2 cell culture supernatants. Anti-angiogenic activity was tested according to a published method [17].

### 3.3. Materials

Samples of *C. crassifolius* were obtained in various areas in China: Guangdong-1 (Guangzhou, China), Guangdong-2 (Foshan, China), Guangdong-3 (Shantou, China), Guangxi-1 (Hezhou, China), Guangxi-2 (Liuzhou, China), Guangxi-3 (Nanning, China), Yunnan-1 (Kunming, China), Yunnan-2 (Dali, China), Hainan-1 (Haikou, China), Hainan-2 (Sanya, China), Guizhou-1 (Guiyang, China) and Guizhou-2 (Bijie, China). Professor Jin-Cai Lu, School of Traditional Chinese MateriaMedica, Shenyang Pharmaceutical University identified the samples. The voucher specimens (No. JGX-20140711) were deposited in the laboratory of Pharmaceutical Analysis, Shenyang Pharmaceutical University.

### 3.4. Extraction Procedures

The roots of *C. crassifolius* (5 g) and solvent were extracted in homogenate extractor (JHBE-50 T, Golden Star Technology, Inc., Ltd. Henan, China) in several HGE conditions. After filtering and transferring the extract to a 100 mL volumetric flask after homogenate treatment, the acetonitrile-water (1:1, *v*/*v*) was added. Before UHPLC-MS analysis, a 0.22-μm membrane filter was used to filter the solution. The single-factor experiments were carried out to systematically study the optimum extraction conditions.

### 3.5. UHPLC and MS Conditions

An Agilent 1290 UHPLC system (Agilent, USA) equipped with a Waters CORTECS C18 column (100 × 2.1 mm, 1.6 μm) was used to separate the nine diterpenoids by chromatography at 30 °C. The flow rate was 0.3 mL/min. A gradient of acetonitrile/formic acid (100:0.1, *v*/*v*, A) and water/formic acid (100:0.1, *v*/*v*, B): 0-8 min, 5−100% A; 8-10 min, 100% A was used to elute the column. The temperature of the autosampler was 4 °C, and the injection volume was 5 μL.

A 6460 QQQ mass spectrometer (Agilent, Santa Clara, CA, USA) was used to implement mass spectrometry through MRM in positive ion electrospray ionization (ESI) mode. The gas temperature was 350 °C, and the capillary was 3.5 kV. The gas pressure of nebulizer was 45 psi. The temperature and gas flow of the sheath were 400 °C and 12 mL/min, respectively. Nitrogen was both nebulizer gas and auxiliary gas. Table 1 shows the optimal precursor-to-product ion pair, fragment (Frag), and collision energy (CE) for each analyte. The data were acquired through Agilent MassHunter Workstation Data Acquisition and processed through Agilent MassHunter Qualitative Analysis (versions B. 06. 00, Santa Clara, CA, USA).

### 3.6. Standard Preparation and Calibration Curves

After preparing standard stock solutions for nine compounds (1 mg/mL) in acetonitrile, they were filtered for analysis using a 0.22-μm syringe filter. The peak area in MRM mode and the analyte concentration was plotted to establish the calibration curve. The solutions were kept at 4 °C.

### 3.7. Method Validation

The reproducibility, recovery, precision, limit of quantification (LOQ), limit of detection (LOD), and calibration curve of the established UHPLC–MS were validated to evaluate the method suitability.

#### 3.7.1. LOQ, LOD, Linearity and Calibration Curve

A sequence of solutions (1–1000 ng/mL of **1**; 5–1400 ng/mL of **2**; 5–1500 ng/mL of **3**; 20–1200 ng/mL of **4**; 5–500 ng/mL of **5**; 2–1200 ng/mL of **6**; 5–2000 ng/mL of **7**; 20–100 ng/mL of **8**, and 2–800 ng/mL of **9**) were obtained by diluting the mixed stock for UHPLC–MS analysis. With analyte concentration as independent variable (*x*-axis) and peak areas as dependent variable (*y*-axis), the calibration curves were established. The correlation coefficient (*r*) of every calibration curve was used to evaluate the linearity. Several decreasing concentrations of analyte were analyzed until the signal-to-noise (S/N). Limit of quantification (LOQ), limit of detection (LOD) at S/N of 3 and 10 were calculated.

#### 3.7.2. Precision

The precision of the established method was evaluated by determining the intra- and inter-day variations. Five successive injection of Guangdong-1 solution on the same day was implemented to evaluate the intra-day precision, and the inter-day precision was measured on five consecutive days. Relative standard deviation (RSD) was the variation.

#### 3.7.3. Accuracy

The accuracy of the method was evaluated by recovery experiments. Guangdong-1 sample was added with nine standards at three different concentrations (0.8, 1.0, and 1.2 times of the concentration in the matrix), and the recovery of each analyte was examined. Every level was analyzed in triplicate. According to the above procedure, the extracted fortified samples were analyzed.

#### 3.7.4. Reproducibility and Stability

The stability of Guangdong-1 solution was evaluated at 4 °C at 2, 4, 8, 24, and 48 h. By comparing the normal concentration and the experimental concentrations, the contents were obtained. The results of six portions of one batch of Guangdong-1 sample indicated the quantitative procedure had reproducibility. The variation was RSD of reproducibility and stability of nine diterpenoids.

### 3.8. Data Analysis

Principal component analysis (PCA) was performed by SIMCA-P 13.0 (Umetrics AB). If the contents of investigated compounds were not detected or lower than the quantitation limit, the values were 0.

## 4. Conclusions

A method with high accuracy and efficiency was proposed to quantify nine anti-angiogenic clerodane diterpenoids in *C. crassifolius*. The method combined HGE and UHPLC-MS techniques was relatively simpler than the traditional analysis methods, which extracted clerodaned diterpenoids in only 5 min and determined the components in 12 min. After validation, the method was successfully applied to determine these diterpenoids in *C. crassifolius* from different sources. Thus, the findings highlighted in the current study are of great significance for establishing a rational QC standard for *C. crassifolius*. Furthermore, the effective method can also be utilized for the extraction and determination of clerodane diterpenoids in other complex plants.

## Figures and Tables

**Figure 1 molecules-24-00694-f001:**
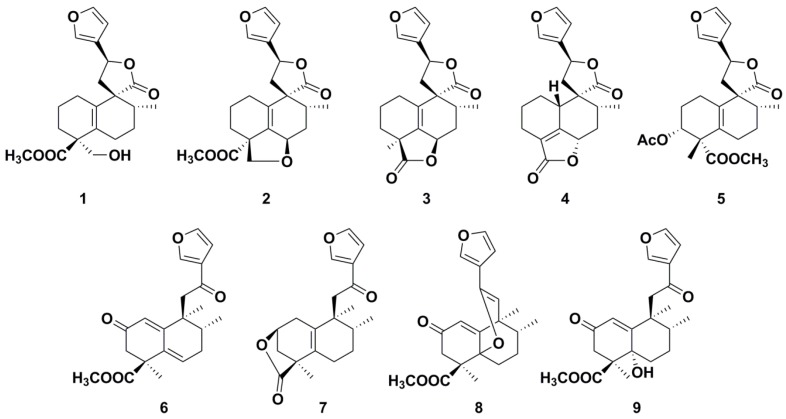
Structures of the nine clerodane diterpenoids.

**Figure 2 molecules-24-00694-f002:**
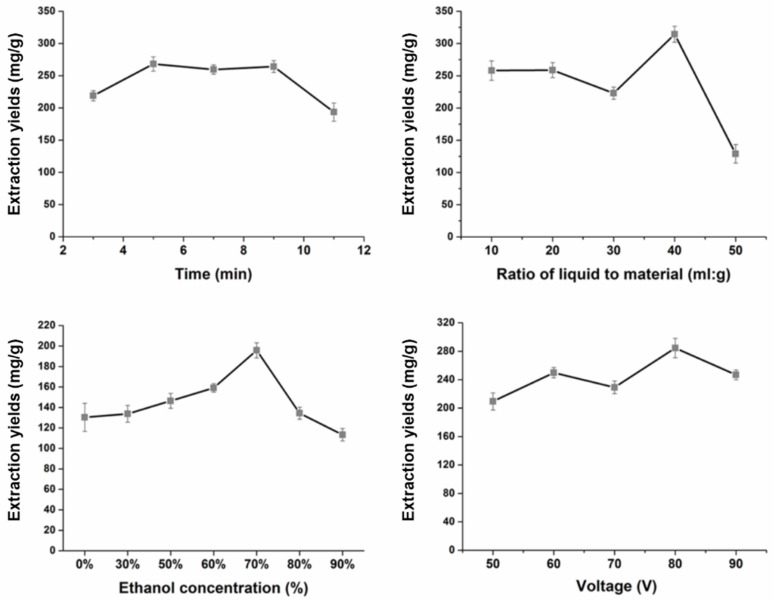
Optimization of homogenate extraction (HGE) procedure (extraction time, ratio of liquid to material, ethanol concentration, and voltage).

**Figure 3 molecules-24-00694-f003:**
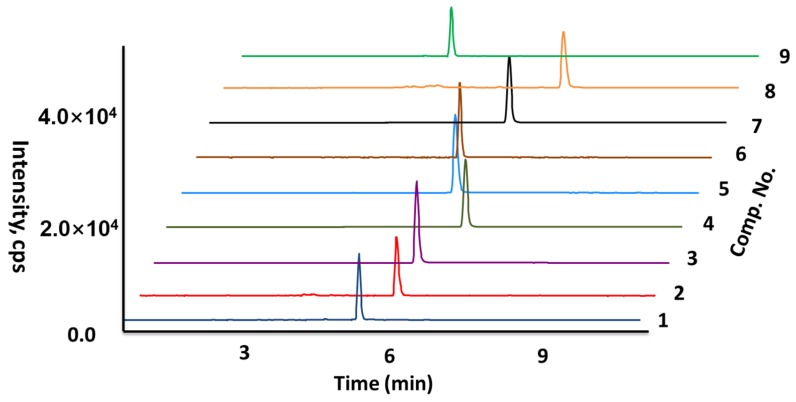
The extracted ion chromatogram of the nine diterpenoids.

**Figure 4 molecules-24-00694-f004:**
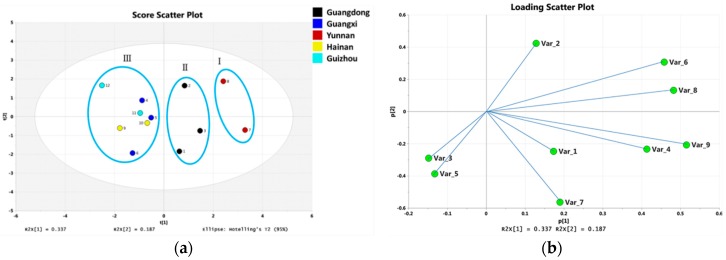
Principal component analysis (PCA)score plot (**a**) and loading plot (**b**) of the 12 batches of *C. crassifolius* samples on the basis of the relative peak areas of 9 diterpenoids.

**Table 1 molecules-24-00694-t001:** Compound-dependent parameters and monitored ion pairs for nine diterpenoids in the multiple reaction monitoring (MRM) mode for UHPLC–MS analysis.

Diterpenoids	Retention Times (min)	Precursor Ion (*m*/*z*) Q1	Produced Ion (*m*/*z*) Q2	Dwell Time (ms)	Frag (V)	CE (V)
**1**	4.861	397.1	367.1	50	154	20
**2**	5.113	390.0	373.1	50	102	4
**3**	5.498	343.0	343.0	50	148	0
**4**	5.622	346.0	175.1	50	102	24
**5**	5.623	439.2	159.1	50	148	32
**6**	5.979	357.0	187.1	50	97	16
**7**	6.108	351.1	241.1	50	220	24
**8**	6.137	379.1	361.1	50	143	16
**9**	6.454	397.2	288.1	50	169	28

**Table 2 molecules-24-00694-t002:** Linear regression parameters, precision, accuracy, reproducibility, stability, and average recoveries of UHPLC–MRM analysis for the nine diterpenoids of *Croton crassifolius*.

Diterpenoids	Linear regression Data			LOD (ng/mL)	LOQ (ng/mL)	Intra-Day Precision (RSD%)	Inter-Day Precision (RSD%)	Reproducibility (RSD%)	Stability (RSD%)	Recovery	
	Linear Range (ng/mL)	Regression Equation	R							Mean (%)	RSD (%)
**1**	10–1000	Y = 1.3629X + 47.268	0.9943	0.363	3.65	3.31	3.65	3.67	3.13	104.5	3.56
**2**	5–1400	Y = 2.3121X + 151.11	0.9961	0.223	3.96	4.54	5.96	4.45	2.78	98.1	3.45
**3**	5–1500	Y = 2.5559X + 43.832	0.9964	0.650	4.21	3.77	4.21	4.18	4.19	103.4	3.98
**4**	20–1200	Y = 2.2157 + 33.717	0.9970	0.924	2.67	2.43	2.67	3.52	2.14	98.2	3.76
**5**	5–500	Y = 2.277X + 15.783	0.9950	0.421	4.78	3.87	4.78	5.71	1.87	98.1	2.97
**6**	2–1200	Y = 2.0473X + 16.295	0.9987	0.0566	0.87	4.34	4.87	4.34	3.96	103.4	2.13
**7**	5–2000	Y = 2.3003X + 18.431	0.9977	0.150	4.45	3.93	4.45	3.14	4.55	102.3	3.67
**8**	20–100	Y = 4.7534X + 101.82	0.9984	0.412	3.31	3.14	3.31	3.65	3.29	101.8	3.33
**9**	2–800	Y = 3.803X + 79.724	0.9989	0.100	1.04	2.44	3.04	3.85	4.21	96.3	4.21

Notes: limit of quantification (LOQ), limit of detection (LOD), relative standard deviations (RSDs).

**Table 3 molecules-24-00694-t003:** The contents of nine diterpenoids in *C. crassifolius* from different sources (*n* = 3, μg/g).

Sources	Diterpenoids									
	1	2	3	4	5	6	7	8	9	Total
Guangdong-1	77.40	330.04	98.54	289,330.11	0.85	38.25	20.20	0.16	145.54	290,041.09
Guangdong-2	77.51	11,075.26	12.07	192,821.96	1.43	361.43	11.35	0.30	120.43	204,481.74
Guangdong-3	102.42	880.56	12.32	276,865.46	1.43	419.17	18.22	0.40	139.01	278,438.99
Guangxi-1	7.63	2,125.32	20.90	112,220.39	1.23	43.70	6.75	0	122.15	114,548.07
Guangxi-2	48.65	112.90	82.50	182,955.12	0	207.63	10.87	0	101.73	183,519.40
Guangxi-3	84.21	9.64	7.85	30,834.32	22.70	0	18.41	0	123.27	31,100.40
Yunnan-1	194.56	236.12	52.87	317,445.29	4.44	442.12	10.30	2.58	199.63	318,587.91
Yunnan -2	46.80	2,959.89	7.76	126,905.53	1.26	905.89	8.57	2.15	157.82	130,995.67
Hainan-1	45.32	1032.11	221.04	145,485.80	7.27	101.74	8.15	0	69.31	146,970.74
Hainan-2	311.18	12.12	65.52	116,355.10	0.75	27.35	6.75	0.23	93.12	116,872.12
Guizhou-1	37.33	652.26	73.58	141,150.43	0.95	10.25	8.15	0.17	107.05	142,040.17
Guizhou-2	21.47	374.45	2.76	57,195.84	2.00	3.97	1.97	0	26.25	57,628.71

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
