# Peer review of "A “Green” Homogenate Extraction Coupled with UHPLC-MS for the Rapid Determination of Diterpenoids in Croton Crassifolius"

_molecules, 2019, doi:10.3390/molecules24040694_

Round 1

Reviewer 1 Report

The manuscript “A “green” homogenate extraction together with UHPLC-MS/MS method to rapid determination of nine major anti-angiogenetic diterpenoids in Croton crassifolius” was submitted to Molecules for publication. The study describes the development of an extraction procedure and the validation of a UHPLC assay for the quality control of C. crassifolius.

Broad comments:

The development of both, the extraction procedure as well as the UHPLC assay are very well described. However, regarding the extraction procedure, the authors should explain how they made sure that they quantitatively extracted the desired compounds.

For the validation of the UHPLC assay, very good results are obtained with respect to precision, limits of detection and quantification. Together with the broad linear range, the method is suitable for the analysis of samples collected in different regions, as they show varying contents of the desired compounds (Table 3). However, determination of accuracy only covered 80 to 120% of the concentration range of one of the samples. Accordingly, samples measured out of this range cannot be considered valid, which is a shortcoming of the presented method. 

Another major issue of the present manuscript is that it contains many mistakes. Not only language errors but also many careless mistakes, making the paper a lot harder to read than it would actually be the case. Some of the mistakes are listed below.

It could also be considered moving the bioactivity results into the supporting information. Though the results are interesting they seem a little disjunct. Even more, as the antiangiogenic effects of Croton crassifolius have been described repeatedly before [1-3].

Summarizing, the method could be a valuable tool for the quality control of Croton crassifolius after validation of a broader concentration range, but the manuscript needs proper revision with regards to the many mistakes before being acceptable for publication.

Specific comments:

The title is grammatically wrong. At least the word “to” has to be changed into “for the” to get it right, but rephrasing of the whole title should be taken into account.

Line 15: The sentence needs to be rephrased.

Line 45 and 46: The sentence is grammatically wrong. “Combined” must be replaced by “combining”.

Figure 3: The title of the y-axis is wrong. Also, “Fixgure.3” in the figure legend is typed wrong. In general, the dots have to be placed behind the numbers.

Line 96: The authors refer to Figure 3 for comparison of peak shapes, but Figure 3 displays the optimization of the HGE procedure.

Line 144: It must read limit “of” detection.

Line 169: compounds 1-9 should be indicated in bold letters.

Lines 180, 188, and…: C. crassifolius must be written in italic letters. Also the plant names in the list of references need to be corrected.

Line 235: …at 4°C_at 2, 4, 8,…

Line 245: The sentence “The optimal homogenate…” must be finished.

Lines 254 and 255: clerodane_diterpenoids

References:

[1] Huang W, Wang J, Liang Y, Ge W, Wang G, Li Y, Chung HY. J Ethnopharmacol. 2015 Dec 4;175:185-91. doi: 10.1016/j.jep.2015.09.021.

[2] Liang Y, Zhang Y, Wang G, Li Y, Huang W. Molecules. 2017 Jan 13;22(1). pii: E126. doi: 10.3390/molecules22010126.

[3] Chen Y, Tian JL, Wu JS, Sun TM, Zhou LN, Song SJ, You S. Fitoterapia. 2017 Apr;118:32-37. doi: 10.1016/j.fitote.2017.02.004.

Author Response

Dear Editors and reviewers

On behalf of my co-authors, thank you very much for giving us an opportunity to revise our manuscript. We appreciate editor and reviewers for the positive comments and suggestions on our manuscript entitled “A “green” homogenate extraction coupled with UHPLC-MS for the rapid determination of diterpenoids in Croton crassifolius”. (ID: molecules-443449). We have studied the comments carefully and have made the corresponding corrections. Revised portion are marked in the paper. We have uploaded the “response to reviewer” in the form of attachments.

Reviewer 2 Report

The introduction needs to be expanded on why the authors were interested in crassifolius in the first place. How and what the authors had hoped this study would contribute to the field?  

I don't understand why the authors mention "environmentally-friendly" to refer to the method? Also what do they mean by "long time consumption" for the traditional methods? Also the dust pollution is not defined. This introduction is not clear and needs a lot of work.

What does HPLC-DAD stand for? All abbreviations must be defined first. 

What does this sentence mean since a PCA cannot be used as quality control: "

The method combined with PCA (principal component analysis) is of great value for 55 the quality control of C. crassifolius from different production regions ". This statement is misleading.

The introduction is so poorly written I couldn't get past it and the rest of the paper doesn't make sense either. Extensive edits are recommended.

Author Response

(The authors gave the same response as above.)

Round 2

Reviewer 1 Report

The manuscript has been revised and most of the points have been clarified / corrected. However, a few spelling mistakes are still to be corrected.

Line 21: "method was only took". The word "was" has to be deleted.

Line 118: "As shown in Table 2" appears twice in a row. The second time it should be either deleted or replaced.

Line 129: "The variations were resulted". The word "were" needs to be deleted.

Line 237: The sentence "The method combined..." doesn't make sense.

Another point is that the authors in section 2.2.2. explain the comparision of three different chromatographic columns and then refer to Figure 3. However, Figure 3 doesn't show a comparison of different stationary phases. Therefore, the reference to Figure 3 must be placed somewhere else. Maybe at the end of the paragraph.

Author Response

On behalf of my co-authors, thank you very much for giving us an opportunity to revise our manuscript (minor revisions)(ID: molecules-443449). We have studied the comments carefully and have made the corresponding corrections. Revised portion are marked in the paper.

Broad comments:

Line 21: "method was only took". The word "was" has to be deleted.

Line 118: "As shown in Table 2" appears twice in a row. The second time it should be either deleted or replaced.

Line 129: "The variations were resulted". The word "were" needs to be deleted.

Line 237: The sentence "The method combined..." doesn't make sense.

Response:

Thank you very much for your thoughtful suggestion. Sorry for our carelessness, we have corrected the grammatical mistakes and deleted the redundant sentences “The method combined HGE and UHPLC-MS techniques was relatively simpler than the traditional analysis methods, which extracted clerodaned diterpenoids only in 5 min and determined the components in 12 min” in the conclusion.

Another point is that the authors in section 2.2.2. explain the comparision of three different chromatographic columns and then refer to Figure 3. However, Figure 3 doesn't show a comparison of different stationary phases. Therefore, the reference to Figure 3 must be placed somewhere else. Maybe at the end of the paragraph.

Response:

Thank you very much for your thoughtful suggestion and we have placed the Figure 3 at the end of the paragraph.

We tried our best to improve the manuscript and made proper changes in the manuscript. These changes will not influence the content and framework of the paper. We appreciate for editor and reviewers’ warm work earnestly, and look forward to hearing from you regarding our submission. We would be glad to respond to any further questions that you may have. Once again, thank you very much for your comments and suggestions.